# Food Desires and Hedonic Discrimination in Virtual Reality Varying in Product–Context Appropriateness among Older Consumers

**DOI:** 10.3390/foods11203228

**Published:** 2022-10-15

**Authors:** Xiao Song, Federico J. A. Pérez-Cueto, Wender L. P. Bredie

**Affiliations:** 1Food Design and Consumer Behaviour, Department of Food Science, Faculty of Science, University of Copenhagen, Rolighedsvej 26, 1958 Frederiksberg C, Denmark; 2Department of Food Nutrition & Culinary Science, Umeå University, 2409 Umeå, Sweden

**Keywords:** virtual reality, immersive technology, context, older consumers, acceptance, protein-enriched food

## Abstract

Immersive virtual reality (VR) videos can replicate complex real-life situations in a systematic, repeatable and versatile manner. New product development trajectories should consider the complexities of daily life eating situations. The creation of immersive contexts of a product with varying levels of appropriateness could be a useful tool for product developers in evaluating the extent to which context may influence food acceptance and eating behavior. This study explored virtual reality (VR) as an efficient context-enhancing technology through evaluations of protein-enriched rye breads and compared the effects of a VR-simulated congruent (VR restaurant) and incongruent (VR cinema) contexts on the acceptance in older consumers. A total of 70 participants were immersed in the two VR contexts and a neutral control context in a randomized order. The responses indicating the desire and liking for rye breads were measured, and the extent of immersion during context exposure was assessed by levels of the sense of presence and engagement. Immersive VR induced positive sensations of presence and a heightened level of engagement. The VR restaurant and neutral contexts were perceived as more appropriate for consuming rye breads and induced higher desire and liking for rye breads, which supported the notion of the alignment of congruent contexts with food desire and liking. The study provides new perspectives, practical methodologies, and discoveries in regard to the creation and application of VR-immersed contexts in food product evaluation. Moreover, it focused on a consumer segment (older consumers) that has seldom been investigated in previous relevant studies. The findings suggest that immersive VR technology, as a tool for evaluating contextual factors, is important for new product development. The good user experience among older consumers further indicated the potential value of VR as a context-enhancing tool for product development.

## 1. Introduction 

Context covers a broad range of phenomena and has been identified as an important factor in models of food choice, food acceptance, and food consumption [1,2,3]. In food sensory and consumer research, the physical condition and social environment, the food or meal, and the individual have been categorized as major variables of context [4]. The congruency or appropriateness of the environment for the food was found to have an assimilation effect, which can influence food acceptance and related behaviors [5,6,7].

Natural or real-world tests and laboratory or other non-natural environment-based tests [1,3] are two of the commonly used approaches for consumer testing. The pros and cons of the two kinds of approaches were explored and compared in earlier studies [2,8,9,10,11]. Real-life contexts (e.g., home use tests) increased consumer engagement and the external test validity compared to lab-based tests. However, the cost of such an approach is usually higher, and the contextual stimuli are more difficult to standardize [2,9,11]. The lab-based approach is timesaving and the test context is more controllable. However, lab-based consumer studies do not account for the influences of natural eating contexts and adequate consumer engagement, which may lower the predictive power of the results compared to a real-world approach [10,12]. 

The immersive technology produces a sense of immersion, which is necessary for evoking feelings of presence and engagement/involvement [13]. Engagement/involvement is a psychological state experienced as a consequence of focusing one’s energy and attention on a coherent set of stimuli or meaningfully related activities and events [13,14]. Through the successful creation of immersive contexts for product testing, the important sources of contextual information in natural settings, i.e., visual, auditory, tactile, and/or olfactory stimuli, can be manipulated and standardized with a greater degree of experimental control and lower execution costs [15,16]. Moreover, immersive technology realizes a repeated simulation of complex real-life contexts at one location and fosters the potential for food behavior, sensory, and consumer evaluations [5]. 

The immersive virtual reality (VR) technology has been reported to produce a greater sense of immersion and improve participants’ feelings of presence and engagement when simulating real-world surroundings [5,17,18,19]. VR makes use of three-dimensional (360°) visual static displays or movies that are either computer simulations of a situation or recordings from real-world events. The immersive state created by VR is further enriched by auditory stimuli [20] using head-mounted displays (HMD) or VR headsets. 

An immersive VR context that produces a greater sense of immersion and presence can produce a higher strength of engagement [13]. A few recent studies investigated the applicability of VR to food-related consumer acceptance and behavior research [5,18,19,21,22,23,24,25,26,27,28,29]. For the exploration of consumer food choices in supermarkets and buffets, researchers created 3D virtual supermarkets and buffets, wherein users could walk around and choose virtual food objects [17,19,27,30,31]. Immersive VR supermarkets tended to depict actual in-store shopper behavior more accurately through the improvement of users’ perceived presence and feelings of being engaged [19]. Andersen et al. (2019) reported that a VR-simulated beach context elicited stronger feelings of presence and engagement and a higher desire for cold beverages compared to the image-immersed beach context. It should be noted that Andersen et al. (2019) evaluated the effects of the retention of contextual exposure on consumer food choices and found that the behavioral impacts of VR post-exposure seemed to be negligible, which indicated that participants could adapt from a VR context to another situation rapidly. These results support the notion that VR has the potential to be applied as a tool for enriching contextual stimuli in sensory and consumer field studies. 

Recently, VR was used to create immersive interactive environments where older consumers could socialize during the consumption of a meal. However, the relevant work was limited in terms of the number of participants and was unable to show that the virtual social interaction significantly influenced food intake [24,32]. Moreover, thus far, no studies have investigated whether immersive VR can influence the hedonic and appetitive responses to food among older consumers. 

In the present study, product–context appropriateness induced by immersive VR and its influence on food desires and hedonic discrimination in the context of real food tasting by older consumers were investigated. This consumer group are advised to increase their protein intake in meals. The product development of protein-enriched products requires studies on preferences in appropriate and versatile contexts. The food stimuli used in this study were protein-enriched rye breads, as part of a typical Danish lunch, which fulfilled the nutritional needs of a meal targeted at older adults [33]. 

The present study aimed to explore VR as a context-enhancing tool in food product evaluations among older consumers. Moreover, the influences of congruent and incongruent contexts on older consumers’ desire and liking for protein-enriched rye breads were compared. In total, three different evaluative contexts were compared: (1) an immersive VR restaurant; (2) an immersive VR cinema; and (3) a default context in a neutral room without exposure to a VR context. Immersive VR contexts were chosen to represent conditions that were congruent (casual lunch restaurant) and incongruent (cinema) with the consumption of rye bread.

## 2. Methods

### 2.1. The Pilot Screening of the Testing Contexts

The screening of a variety of food contexts was conducted so as to select congruent and incongruent usage contexts of consuming rye bread. Eleven usage contexts were depicted and presented with corresponding images for visual elicitation. To measure the perceived appropriateness of each context for eating rye bread, the following question was asked: “How likely is it that you will eat rye bread in the following situations?” Participants rated the appropriateness of each context with on a 7-point Likert scale (1 = extremely unlikely, 4 = neither likely nor unlikely, 7 = extremely likely). Samples of the images used are shown in Figure 1. The following eleven contexts were included in the screening test: eating alone at home; picnic in a natural environment; while on transportation vehicles; at a luxurious restaurant; at a casual restaurant; at home with family or friends; at a stadium; at a cinema; at a canteen; as foods during a break from, or after, indoor sports; as foods during a break from, or after, outdoor sports.

In total, 72 participants (61% females) aged between 61 to 83 years participated in the screening test. They were recruited from the external consumer panel of the Department of Food Science, University of Copenhagen.

### 2.2. Study Design

#### 2.2.1. Contexts

Based on the results of the screening of the contexts, the casual restaurant context was regarded as the most congruent for consuming rye bread and was chosen as the representative congruent context. The cinema context rated as the least congruent context and was chosen to represent the incongruent context for eating rye bread. 

In addition to an immersive virtual reality restaurant context (VR restaurant) and an immersive virtual reality cinema context (VR cinema), a default ambient room context (neutral) was included in the study. Participants experienced all three contexts in the neutral room of The Future Consumer Lab at the Department of Food Science, University of Copenhagen.

The VR restaurant context was presented through a 7 min 360° film shot at a casual restaurant in Copenhagen, Denmark, using an Insta360 Pro Spherical VR camera (Shenzhen Arashi Vision Co., Ltd., Shenzhen, China). The sound of the restaurant, e.g., the background music, was recorded as well. The VR cinema context was presented by a 360° 7 min movie comprised of three movie trailers played through the Oculus VR Cinema application on a Samsung Galaxy S7 cellphone. During exposure to the VR contexts, participants watched the film using the VR headset (a Samsung Gear VR combined with a Samsung Galaxy S7 cellphone) and the audio headset (Beyerdynamic DT-880) (Figure 2). The default environment was controlled for temperature (22 °C) and white overhead lighting and was furnished with a dining table and chair. Figure 3 shows the restaurant environment and the cinema environment.

#### 2.2.2. Procedure

The study procedure is demonstrated in Figure 4. During the test, the participant sat in the dining chair beside the dining table in the neutral room.

VR was used as a context-enhancing technology for the evaluation of protein-enriched rye breads. Each participant experienced all three contexts in a randomized order. In the beginning of this section, an interviewer gave a briefing of the test and an introduction explaining how to wear and adjust the VR headset. After that, the participants were given around 1 min to adjust to the VR and audio headset and visually explore the environment, with a soundscape to develop a sense of place identity and presence [24,34]. After the participants had formed a sense of the VR environmental representation, the participants’ engagement was measured, and then three rye bread samples were tasted and evaluated through interviews, with water provided for mouth cleansing between samples. The bread samples and water were handed to participants by the interviewer while the participants were experiencing the VR contexts with the VR headset on. At the end of each VR context exposure, participants were interviewed about feelings of dizziness and usage-related implications while wearing and taking off the VR headset, followed by a 2 min break before experiencing the next context. The same procedure was followed during exposure to the neutral context, although questionnaires instead of interviews were used to collect data on the participants’ engagement and desire and liking for bread. The order of rye bread samples was randomized and balanced. The whole session lasted around 25 min in total.

### 2.3. Rye Bread Samples

All three rye bread samples were evaluated during exposure to each context, which included one non-enriched control sample, one 7%-whey-protein-isolate-enriched sample (pro 1), and one blend-protein (4% soy protein isolate + 4% whey protein hydrolysate)-enriched sample (pro 2). Details of the rye bread recipes and processing procedures have been reported elsewhere (Song et al., 2018). Breads were baked the day before each test day and then preserved and served at room temperature (22 °C). Bread samples were cut into 0.85 cm-thick and approximately 2.0 cm × 2.0 cm-sized cubes and put into 60 mL-sized sample cups with lids, labeled with three-digit codes. Each cup contained two pieces of bread cubes, and each cube had one side of crust.

### 2.4. Participants

A total of 70 Danish older adults (44 females) aged between 60 to 80 years participated in the study. They were recruited through the external consumer database of the Department of Food Science, University of Copenhagen. All participants signed informed consent to participate in the study before the test. The interviewer gave a clear explanation of the whole procedure at the beginning of the study, with an information sheet provided to each participant. Participants were instructed to refrain from eating and drinking coffee or tea for 1.5 h prior to the starting time of the test. A six-digit identification code was assigned to each participant. Data were collected and handled following the General Data Protection Regulation (GDPR) in EU law.

### 2.5. Measurements

#### 2.5.1. Background Information

Demographic information including age, gender, height, weight, self-reported health status, exercise frequency, education level, and living status were collected. Visceral hunger and satiety were measured on a 150 mm visual analogue scale (VAS) at the beginning of each context exposure. 

#### 2.5.2. Self-Reported VR Experience

The familiarity with the VR headset was assessed on a 7-point scale ranging from “−3 = had never heard about it” to “+3 = being extremely familiar”. Participants were asked to state whether they felt dizzy or experienced any other usage-related implications during each immersive VR contextual exposure.

#### 2.5.3. Presence, Engagement, and Perceived Context–Product Appropriateness

The feeling of presence during the VR contextual exposures was evaluated using the terms of “feels like being in a restaurant” and “feels like being at a cinema” on a 7-point agreement Likert scale with −3 = strongly disagree, 0 = neither agree nor disagree, and +3 = strongly agree. The perceived context-induced appropriateness of all three contexts was evaluated using the terms “the context is appropriate for eating rye bread” on the same scale. The participants’ liking for the context was rated on a 9-point scale, with −4 = extremely dislike, 0 = neither like nor dislike, and +4 = extremely like.

During each context exposure, the levels of feeling excited, comfortable, happy, and relaxed were rated on a 7-point Likert scale, with −3 = extremely unexcited/extremely uncomfortable/extremely unhappy/extremely not relaxed and +3 = extremely excited/extremely comfortable/extremely happy/extremely relaxed. Participants could report other feelings that they experienced during each contextual exposure.

#### 2.5.4. Desire and Liking for Rye Breads

The desire for rye bread was measured before the tasting of the samples during each context exposure on a 7-point Likert scale, with −3 = extremely undesirable and +3 = extremely desirable. The overall liking for each rye bread sample was measured on a 9-point hedonic scale, with −4 = dislike extremely, 0 = neither like nor dislike, and +4 = like extremely.

### 2.6. Data Analysis

The SPSS Statistics 25 (IBM, Armonk, NY, USA) software package was used for the data analysis. One-way analysis of variance (ANOVA) followed by Fischer’s least significant difference (LSD) post hoc test was performed to assess differences in product–context appropriateness, feelings of presence and engagement, and desire and liking for rye breads between and within contexts. Pearson’s correlation coefficients were used to investigate interrelations between product–context appropriateness, feelings of presence and engagement, and the desire and liking for rye breads. For all analyses, *p*-values of less than 0.05 were considered statistically significant. 

## 3. Results 

### 3.1. Demographic Characters and Self-Reported VR Experience

The demographic characteristics and background information of the participants are shown in Appendix A. The average level of familiarity with the VR headset was 2.8 ± 1.3 (mean ± SD), corresponding to the ‘somewhat familiar’ level. Males were found to be more familiar with VR headsets than female participants (*p* < 0.05).

Up to 7.1% and 4.3% of the participants reported dizziness in the VR contexts, respectively. Up to 47.1% and 42.9% of the participants reported usage-related implications during VR restaurant and VR cinema exposure due to one or more of the following statements: own glasses, pressure on the nose, the weight of headset, and blurry film. The blurry film was corrected by adjusting the fit of the head-mounted display during the adjustment period at the beginning of each VR contextual exposure.

### 3.2. Context–Product Appropriateness and Participants’ Feelings of Presence and Engagement

The participants’ feelings of presence, defined as “feels like in a restaurant” or “feels like in a cinema”, were positive for both the VR restaurant and the VR cinema. The VR restaurant induced the strongest engagement levels of feeling excited, comfortable, happy, and relaxed among the three contexts. The VR cinema and neutral context showed no significant differences in their ability to evoke excitement comfortable, happy, and relaxed feelings (*p* > 0.05). The VR restaurant immersion elicited significantly a higher level of excitement than the neutral context and the VR cinema context immersion (*p* < 0.05).

The results of perceived context–product appropriateness, the perceived liking for the context, and participants’ feelings of presence and engagement are shown as scores in Table 1. The VR-cinema context was perceived as very incongruent (−2.2) for eating rye bread. The VR restaurant and neutral contexts were regarded as congruent with the consumption of rye bread, which had significantly higher appropriateness scores compared to VR cinema (*p* < 0.05).

A Pearson’s correlation analysis was conducted to evaluate the interrelations (1) between the perceived context–product appropriateness and participants’ feelings of engagement; (2) between the perceived liking for the context and participants’ feelings of engagement; and (3) between participants’ feelings of presence in the VR contexts and feelings of engagement. The perceived appropriateness showed positive correlations with the level of feeling comfortable (r = 0.24, *p* < 0.01), level of feeling excited (r = 0.16, *p* < 0.05) and level of feeling happy (r = 0.17, *p* < 0.05). Similarly, the perceived liking for the context also showed positive correlations with the level of feeling comfortable (r = 0.39, *p* < 0.01), level of feeling excited (r = 0.36, *p* < 0.01), level of feeling happy (r = 0.30, *p* < 0.01), and level of feeling relaxed (r = 0.32, *p* < 0.01). In the VR contexts, weak positive correlations were found between the feelings of presence and the level of feeling excited (r = 0.35, *p* < 0.01), the level of feeling comfortable (r = 0.34, *p* < 0.01), the level of feeling happy (r = 0.27, *p* < 0.01) and the leveling of feel relaxed (r = 0.25, *p* < 0.01). Thus, the more congruent the context was with the consumption the rye bread, the more comfortable, excited, and happy participants felt. Moreover, the feelings of presence were consistent with all four feelings of engagement.

### 3.3. Participants’ Stated Desire and Liking for Rye Bread

The results of the tests of participants’ stated desire for rye breads across the three contexts are compared in Figure 5A. The comparisons of the participants’ liking for rye breads between and within contexts are shown in Figure 5B and Figure 5C, respectively.

Participants’ liking for all three rye breads (control, pro 1, pro 2) were lowest while exposed to the VR cinema context (Figure 5B). Compared with the VR cinema, consumer liking for pro 1 bread and pro 2 bread was significantly higher in the VR restaurant context (*p* < 0.05), and participants’ liking for control bread and pro 2 bread was significantly higher in the neutral context (*p* < 0.05). 

The hedonic discrimination of the rye breads was only possible in the neutral context, whereas the consumers’ liking for the products could not be discriminated in the VR contexts (Figure 5C). A Pearson’s correlation analysis was conducted to evaluate the interrelations between the participants’ desire for rye bread, perceived product–context appropriateness, perceived liking for the context, feelings of presence and engagement, and liking for rye breads. The average values of consumers’ liking for the three rye bread samples in each context were used for the correlation analysis. The results showed that participants’ desire for rye bread was positively correlated with perceived product–context appropriateness (r = 0.52, *p* < 0.01), perceived liking for the context (r = 0.15, *p* < 0.05), liking for the rye bread (r = 0.21, *p* < 0.01), and engagement measurements, which included feeling comfortable (r = 0.23, *p* < 0.01), feeling excited (r = 0.25, *p* < 0.01), feeling happy (r = 0.27, *p* < 0.01) and feeling relaxed (r = 0.17, *p* < 0.05). Participants’ liking for the rye bread was positively correlated with the level of product–context appropriateness (r = 0.20, *p* < 0.01), perceived liking for the context (r = 0.20, *p* < 0.01), level of feelings of presence (in VR-contexts; r = 0.23, *p* < 0.01), level of feeling excited (r = 0.15, *p* < 0.05), and desire for the rye bread (r = 0.21, *p* < 0.01). 

The results were compared across the segments of consumers with different backgrounds, and no significant differences in the context–product appropriateness, feelings of presence and engagement, and desire and liking for rye breads were found.

## 4. Discussion

Immersion is an important construct to consider when investigating the impacts of mediated experiences on consumer cognition [35]. This study evaluated immersion through the measurement of presence sensation and engagement. The results indicated that the virtual reality experience to some degree could represent the sensation of a real-life situation and gave a heightened level of engagement among older participants. This finding is in line with earlier research and further demonstrates that the involvement of multi-sensory cues in a VR-simulated context can improve users’ presence sensation and engagement [5,18,19,27,28]. 

When experiencing the VR restaurant context and neutral context, participants perceived a higher level of contextual appropriateness and acceptance for consuming rye breads in contrast to the VR cinema context. Furthermore, participants’ desire and liking for rye breads were significantly correlated with the level of product–context appropriateness and contextual acceptance, which is in line with previous studies that demonstrated the assimilation effects of the situational appropriateness and acceptance of the product [6,36]. 

The discrimination efficacy with respect to the liking for the rye bread samples was reduced in both of the immersive VR contexts in contrast to the neutral context. In previous studies, no consistent trend regarding the effects of the evoked contexts on hedonic discrimination was reported [37]. Thus far, the reasons for these inconsistencies are not clear. The magnitude of the product differences might be one reason in some studies [38,39,40]. In the current study, it could be hypothesized that, if the bread sample hedonic differences were larger, the contextual effect on hedonic discrimination would be less likely to have an effect. Hersleth et al. [41] reported greater hedonic discrimination when consumers evaluating dry-cured ham samples were presented with a ‘traditional meal’ image than with a ‘novel meal’ image. This finding indicated that consumers’ associations with the ‘novel meal’ situation decreased their hedonic discrimination, possibly due to the higher attentional load of the ‘novel meal’ situation. Although the immersive contexts led to a difference in the hedonic ratings of the samples, with higher hedonic ratings in the restaurant setting, the level of hedonic discrimination in both immersive contexts was lower when compared to the neutral setting. Participants were more engaged when experiencing the VR contexts, which might have weakened their attention in discriminating the hedonic differences between the food samples, compared to the neutral context. The discrimination efficacy in VR-simulated contexts might be improved by merging the VR environment with actual interactions between users and foods in real time [24]. 

In this study, consumers’ desire for food was strongly linked to the appropriateness of the eating context, which is in line with previous studies [1,5] and further proves that desire is context-dependent. Moreover, it was found that a positive association with the eating context increases the liking for a food to some extent. The degrees of food desire and liking are known factors involved in determining choice, appetite, and food consumption. Therefore, the results indicate that eating contexts that are inappropriate to the meal should be omitted when aiming to stimulate food consumption in older people. Furthermore, the marginal positive effects on appetite may be expected when optimizing the physical eating context [1,42,43,44,45,46]. 

The interview approach was chosen because older participants might feel uncomfortable or unconfident using digital scales by themselves in order to give ratings of their engagement and overall liking. However, participants may tend to choose reasonable answers to please the interviewer [47]. A comparison between the influences of these data-collecting approaches could be of interest in future studies. Moreover, in the interest of ensuring result reliability, Likert scales with fewer points were chosen to render the scales easier for older participants to use; 9-point scales were applied to measure consumer liking and 7-point scales were chosen for the measurements of feelings [42,44,48].

The usage-related problems and the dizziness reported by some participants in this study had no significant influences on the participants’ engagement and liking for the rye breads. Similar usage-related problems were found in a few recent VR studies as well [5,28]. When designing VR studies, these problems should be taken into consideration, and counteracting strategies require further exploration in order to improve user experience.

The food acceptance evaluation in this study has some limitations, as the VR headset shields the participants from seeing the real food stimuli during the exposure to VR-simulated contexts. Participants could see the bread sample before they put on the VR headset, and when they had the VR headset on, they were able to hold the sample cup, which the interviewer handed to them, and pinch and bring the bread sample to their mouths by themselves. However, such an interaction with food samples is unnatural and restrictive. The application of a mixed-reality-simulated context which merges the VR-immersed environment with the natural interaction between users and real-world elements, such as hands and foods, enhances the natural interactions in the virtual context [24,49]. Moreover, this augmented reality might be another option that can be used to enable the users to see the real food product, since it can transfer the virtual environmental information into the user’ real-life environment in real time [50]. 

Social interactions, which play important roles in influencing consumer food-related behavior, are absent in most VR studies. Through virtual avatars, VR could enable users to communicate and eat with others virtually [51,52]. Further integration of social interactions into the virtual world will create a more complete setup for relevant consumer behavior studies. Moreover, for older adults, who are more likely to eat meals alone [53], the technique could be applied to make meals more enjoyable and appetizing and may thus contribute to the improvement in the older adults’ food intake and nutritional status [54,55,56]. 

## 5. Conclusions

This study simulated congruent and incongruent eating environments through 360° movies in mounted VR headsets and assessed their effects on food desire and liking by older consumers. Immersive VR induced a positive sensation of presence and enabled contextual manipulations through which older people were more engaged. The incongruent product context led to a corresponding decrease in both desire and liking among older consumers. The immersion of older consumers in both the appropriate and less appropriate virtual eating contexts reduced the level of hedonic discrimination between protein-enriched foods. In the neutral laboratory setting, the protein-enriched foods could be discriminated based on the consumers’ liking for each. These findings suggest that, considering the complexities of daily life eating situations, the creation of immersive contexts with varying levels of appropriateness for a product could be an efficient and valuable tool for product developers in evaluating the extent to which context may influence food acceptance and eating behavior. The use of immersive VR technology for evaluating contextual factors is important for future food product development. The natural interactions in the virtual context could be further enhanced, using augmented reality, to merge the VR-immersed environment with interactions between users and real-world elements.

## Figures and Tables

**Figure 1 foods-11-03228-f001:**
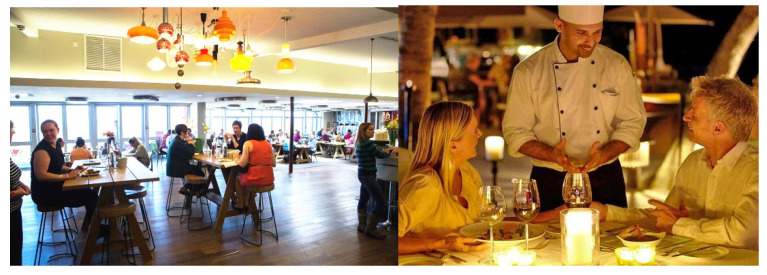
Example images used for the visual elicitation of contexts in the screening test. Left: a casual restaurant; right: a luxurious restaurant.

**Figure 2 foods-11-03228-f002:**
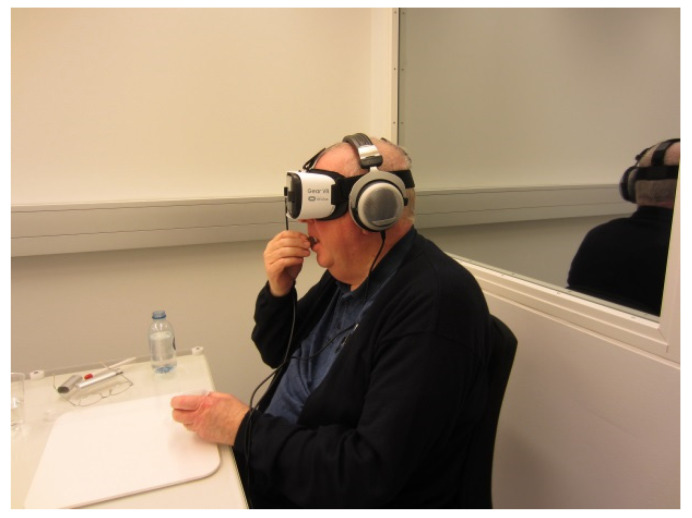
Photo of a participant tasting rye bread during VR contextual exposure.

**Figure 3 foods-11-03228-f003:**
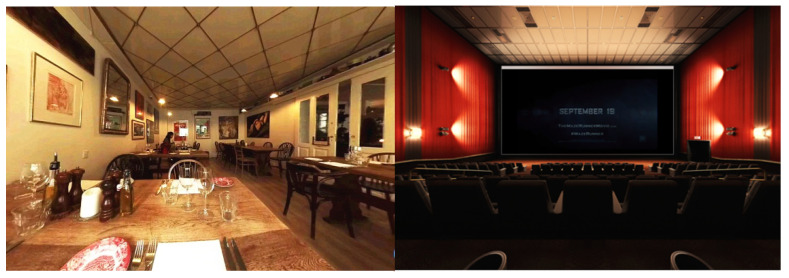
The casual restaurant context (**left**) and the cinema context (**right**).

**Figure 4 foods-11-03228-f004:**
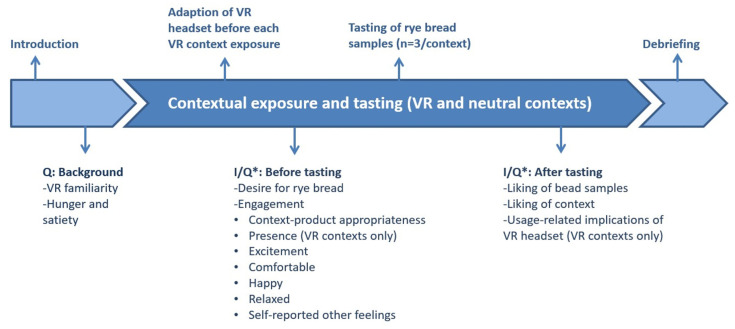
Illustration of the sequence of measuring responses in the given context. Each participant experienced three contexts and tasted rye bread samples in each context. Q = self-reported questionnaire; I/Q* = interview during VR context exposure and self-reported questionnaire used during the neutral context; VR = virtual-reality-immersed restaurant context and cinema context.

**Figure 5 foods-11-03228-f005:**
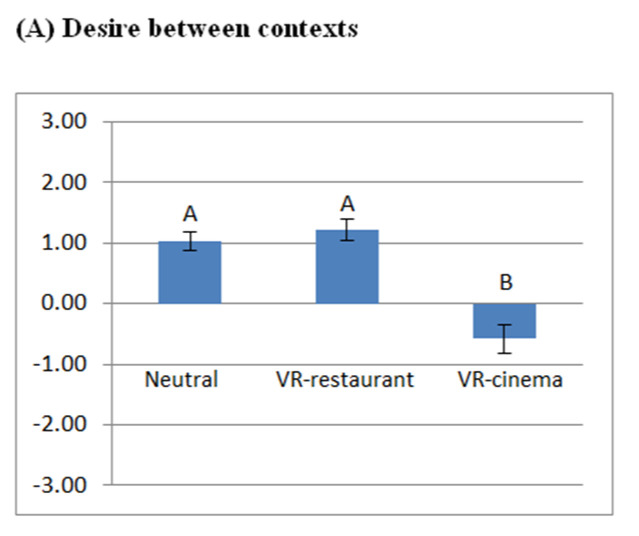
Participants’ stated desire (mean ± SEM) for the rye bread across three contexts (**A**) and comparisons of liking scores (mean ± SEM) for rye breads between (**B**) and within contexts (**C**). Different letters indicate significant differences at *p* < 0.05. NS = non-significant. Control bread = rye bread without protein enrichment. Pro 1 bread = rye bread enriched with 7% whey protein isolate. Pro 2 bread = rye bread enriched with 4% soy protein isolate + 4% whey protein hydrolysate.

**Table 1 foods-11-03228-t001:** Comparison of context–product appropriateness and participants’ presence sensation and engagement between contexts (n = 70).

	Neutral	VR Restaurant	VR Cinema
Context–product appropriateness	0.7 ^A^	0.8 ^A^	−2.2 ^B^
Liking of context	1.6 ^AB^	2.0 ^A^	1.2 ^B^
Presence and engagement			
Feels like being in a restaurant/cinema	N/A	1.3	1.8
Feel excited	1.0 ^B^	1.5 ^A^	1.1 ^B^
Feel comfortable	1.6 ^B^	2.0 ^A^	1.7 ^AB^
Feel happy	1.8	2.2	1.9
Feel relaxed	1.9	2.2	2.0

Notes: The perceived context-induced appropriateness and feelings of presence during the VR contextual exposures were evaluated using the terms “the context is appropriate for eating rye bread”, “feels like being in a restaurant” and “feels like being at a cinema” on the 7-point Likert scale of agreement, with −3 = strongly disagree and +3 = strongly agree. The liking for the context was rated on a 9-point scale, with −4 = extremely dislike, 0 = neither like nor dislike, and +4 = extremely like. The levels of feeling excited, comfortable, happy, and relaxed were rated on a 7-point Likert scale, with −3 = extremely unexcited/uncomfortable/unhappy/not relaxed and +3 = extremely excited/comfortable/happy/relaxed. Different letters indicate significant differences at *p* < 0.05.

## Data Availability

The data that support the findings of this study are available on request from the corresponding author. The data are not publicly available due to the containing of information that could compromise the privacy of research participants.

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
