# Peer review of "Food Desires and Hedonic Discrimination in Virtual Reality Varying in Product–Context Appropriateness among Older Consumers"

_foods, 2022, doi:10.3390/foods11203228_

Round 1

Reviewer 1 Report

Some suggestions.

The subject is very innovative. The manuscript is written well with explanations of the design of the experiment and presenting and discussing results. The question is, what is new contributed to the food science discipline?

The research group is seniors, and we can learn their food desire and likes depending on simulated congruent and incongruent eating environments by 360o movies in mounted VR headsets. Moreover, the immersion of older consumers in both the appropriate and less appropriate virtual eating contexts reduced the hedonic discrimination among protein-enriched foods. The product development of protein-enriched foods for older people should consider the complexities in daily-life eating situations. That immersive VR technology seems to be a promising and efficient tool for evaluating contextual factors in food acceptance research for older consumers.

The abstract states that new product development should consider the complexities of daily-life eating situations. And that creating immersive contexts with varying appropriateness for a product could be a valuable tool for product developers to evaluate the extent to which context may influence food acceptance and eating behavior.

Maybe it is necessary to connect both parts of the manuscript (abstract and conclusions) to emphasize that this preliminary and novel idea is of future importance, not only with attitude to seniors to food. Consider also the connection with the text from lines 360-368.

Moreover, there is no discussed connection between background information (demographic information of age, gender, height, weight, self-reported health status, exercise frequency, education level, and living status) and obtained results.

Author Response

Thank you for your comments. Please see attachment for the authors' responses.

Reviewer 2 Report

The Authors have chosen a topic of emerging interest, the effect of virtual reality (VR) on product preference or product acceptance. In the Introduction part, they give a broad overview of that field, listing the most relevant and recent references related to the topic.

The pre-screening of the possible VR context is well established, the test protocol is clearly shown at Figure 4. The randomization of the scenarios is a good approach to balance any undesirable effect from the experimental plan.

At Line 193, we read that the participants have received cubes (2 cm x 2 cm) from the bread. Since this research is more focused on, the VR factor I do not think that this is an important question, but in most cases, we give consumers samples usually the way they consume that. In case of bread that would mean slices. We also know that sometimes sample quantity can be a limiting factor, or the sample holder can limit the sample size. However, if you plan to do further studies with solid food samples I would encourage you to use sample servings, which are true ‘pictures’ of the way we consume them usually.

At section 2.5.3. we read that most Likert scales were 7 points, except the liking of the context which was measured on a 9 point scale. According to my experience in preference studies the 9 point scale is generally accepted and used. I would advise to insert a sentence about, why researcher have chosen the less distinguished 7 point scale for the measurement of other factors.

Figure 6, section C teaches us a very important message: if consumers evaluate the samples in an immersive environment, the differences in product preference disappear (there were no significant differences). However, an interesting issue would be to investigate the possible heterogeneity of the responses by the application of cluster-analysis. I am aware of the fact that the sample size in the current study would cause small sub-groups in cluster analysis, so probably that dataset is not suitable for that purpose. So it is not necessary to add a section of cluster-analysis of the current manuscript.

It would be interesting to know, what kind of sensory attributes would make the difference between samples in the study. Were they related to texture, flavor or odor descriptors, or both?

In the Discussion section, the Authors also mention the possibility of augmented reality, which is an important and promising technique to integrate sample related and environment related factors.

Generally the study is well-established and interesting work.

Technical note: At the acknowledgement section the link (calm.ku.dk) did not work.

Author Response

Thank you for your nice comments. Please see the attachment for our responses. 

Reviewer 3 Report

The manuscript is very original and topical, as it explores virtual reality (VR) as an  context-enhancing technology, in evaluating protein-enriched rye breads, in congruent contexts (VR-restaurant) and incongruent contexts (VR-cinema) on acceptance by older consumers. The results suggest that product development of protein-enriched foods for older people should consider the complexities of daily life food situations.

The paper is well written and clearly illustrates  the intent and conclusions of the researchers

Author Response

Thank you for your nice comments!

Round 2

Reviewer 1 Report

I appreciate your efforts to make your manuscript more valuable and high-quality. I am satisfied with your explanations. 

Reviewer 2 Report

After the first revision, in this second round I accept to present this paper.